# Novel Gene-Informed Regional Brain Targets for Clinical Screening for Major Depression

**DOI:** 10.3390/neurolint17060096

**Published:** 2025-06-19

**Authors:** G. Lorenzo Odierna, Christopher F. Sharpley, Vicki Bitsika, Ian D. Evans, Kirstan A. Vessey

**Affiliations:** Brain-Behaviour Research Group, University of New England, Armidale, NSW 2351, Australia; godierna@une.edu.au (G.L.O.); vicki.bitsika@une.edu.au (V.B.); ievans3@une.edu.au (I.D.E.); kvessey@une.edu.au (K.A.V.)

**Keywords:** depression, Parkinson’s disease, temporal pole, Heschl’s gyrus, hippocampus, perirhinal gyrus, ventral tegmental area, postcentral gyrus, olfactory tubercle, insular cortex

## Abstract

**Background/Objectives:** Major Depression (MD) is a common disorder that has significant social and economic impacts. Approximately 30% of all MD patients are refractory to common treatments, representing a major obstacle to managing the impacts of depression. One potential explanation for the incomplete treatment efficacy in MD is a substantial divergence in the mechanisms and brain networks involved in different subtypes of the disorder. The aim of this study was to identify novel brain regional targets for MD clinical screening using a gene-informed approach. **Methods:** A new analysis pipeline, called “Analysis Tool for Local Association of Neuronal Transcript Expression” (ATLANTE), was generated and validated. The pipeline identifies brain regions based on the shared high expression of user-generated gene lists; in this study, the pipeline was applied to discover brain regions that may be significant to MD. **Results:** Nine discrete brain regions of interest to MD were identified, including the temporal pole, anterior transverse temporal gyrus (Heschl’s gyrus), olfactory tubercle, ventral tegmental area, postcentral gyrus, CA1 of the hippocampus, olfactory area, perirhinal gyrus, and posterior insular cortex. The application of network and clustering analyses identified genes of special importance, including, most notably, PRKN. **Conclusions:** This study provides two major insights. The first is that several brain regions have unique MD-associated genetic architectures, indicating a potential explanation for subtype-specific dysfunction. The second insight is that the PRKN gene, which is strongly associated with Parkinson’s disease, is a key player amongst the MD-associated genes. These findings reveal novel targets for the clinical screening of depression and reinforce a mechanistic connection between MD and Parkinson’s disease.

## 1. Introduction

Major Depressive Disorder (MDD) [1] is indicated by the presence of at least five of nine diagnostic criteria, including a persistent depressed mood, loss of interest or pleasure in previously enjoyable activities, recurrent thoughts of death, and physical and cognitive symptoms [2]. MDD is a leading cause of disability, and a major contributor to the global burden of disease [3]. Experienced by over 280 million individuals globally, MDD has wide-ranging effects on physical health [4,5,6], cognitive abilities [7], social relationships [8], functional capacity [9], and quality of life [10,11]. Although MDD diagnosed by clinical interview is the gold standard, sometimes other methods are used to identify the presence of the MDD diagnostic criteria, such as patient self-report on standardised questionnaires. These diagnoses are sometimes referred to as MDD, Major Depression (MD), or Depressive Disorder (DD) [12], and depend upon the MDD-related validity of the self-report scale items. For example, Beck’s Depression Inventory [13] (a very common measure of MDD in trial research) does not include all the diagnostic criteria for MDD, but other self-report scales, such as the PHQ-9 [14] and the SDS [15], do, and demonstrate significant agreement with clinician interviews, a finding consistent with the results from a comparison of other self-report scales and clinician interviews. Although not directly relevant to the current study, relatively fewer of the diagnostic criteria are associated with similar deleterious effects on daily functioning. For example, Sub-Syndromal Depression (SSD), which requires only two of the diagnostic criteria to be present, has been reported to show no large consistent differences in impairment to MDD patients [16], and leads to the same medical burden as that of MDD patients [17]. Thus, because various studies reviewed below identified their participants via clinician interviews or self-report scales, the current study collapsed MDD, MD, and DD into the category of ‘Major Depression’ (MD) to report on the genetic associations.

The prevalence and broad impacts of MD argue for the development of models that can inform clinical practice. One of the major underlying influences of the prevalence and severity of MD is genetic factors, and particularly their interaction with environmental factors. Twin studies have demonstrated that MD has a strong familial component, with most heritability estimates ranging from 30 to 50% [18,19,20]. Despite some earlier claims regarding the identification of ‘depression genes’, a mega-analysis using the genome-wide association study (GWAS) in 2013, featuring the largest cohort at the time (9240 cases), was not able to confidently identify any genetic polymorphisms associated with MD [21]. However, since then, studies using progressively larger or more clinically homogenous cohorts have reported increasing success [22,23,24,25,26,27,28,29,30]. The first to do so was a 2016 study by the CONVERGE consortium, which identified two MD-associated risk loci from a cohort of 5303 Han Chinese women, most of whom met the DSM-IV [31] criteria for MD [28]. By contrast, the most recent study, published in 2025 by the Major Depressive Disorder Working Group of the Psychiatric Genomics Consortium, included 688,808 individuals with MD across 29 countries, and confidently identified a total of 697 risk loci [32]. Combining precise mapping tools with additional approaches, such as expression and protein quantitative trait loci, or Hi-C, has enhanced the translation of single-nucleotide variants to putative genes of interest [33,34,35]. Today, there are hundreds of MD-associated protein coding genes, pseudogenes, and non-protein coding sequences, which together provide a wealth of information on the underlying genetic architecture of MD.

One possible application of this genetic data is to generate testable hypotheses pertaining to the role of specific brain regions in MD. If genes associated with MD share high expression in any one brain region, for example, this may indicate that the brain region of interest is important to the pathogenesis or clinical progression of MD. Such a gene-informed approach might help prioritise brain regions for study, which is valuable when considering the many possible measurable changes to brain structure and function that can occur in people with MD [36,37]. The goal of this study was to use recently identified high-confidence MD-associated genes to identify brain regions that might be fundamental to MD or any of its subtypes. The identification of such brain regions might provide novel high priority targets for the clinical screening of MD subtypes using readily available tools such as electroencephalogram recordings, transcranial magnetic stimulation, or emerging wet biomarker detection toolkits.

## 2. Materials and Methods

### 2.1. Curation of Gene Lists

Genes of interest to MD were sourced from four recent genome-wide association meta-analyses: Levey et al. (2021; [29] N = 1,342,778), Als et al. (2023 [30] N = 1,349,887), Meng et al. (2024 [22] N = 1,815,091), and Adams et al. (2025 [32] N = 5,053,033). These studies were selected because they analysed large cohorts with broad geographical coverage and integrated multiple analytical tools to discover high-confidence single-nucleotide polymorphisms. High-confidence genes from Levey et al. (2021) [29] and Als et al. (2023) [30] were classified as those identified via colocalisation analyses associating expression quantitative trait loci and GWAS signatures. High-confidence genes from Meng et al. (2024) [22] and Adams et al. (2025) [32] were classified as those which were identified via convergence of transcript-wide association, functional mapping, and MAGMA/HiC-MAGMA. High-confidence genes that were identified using only PsyOPS or protein-wide association methods by Adams et al. (2025) [32] were excluded to improve consistency across all studies. Non-protein coding transcripts were excluded from the results of all studies, including long non-coding RNA and pseudogenes. The resultant list used for this study included 268 MD-associated protein-coding genes (Table A1).

‘Positive control’ gene lists were sourced via the BioGPS Human Cell Type and Tissue Gene Expression Profiles dataset [38,39,40]. Highly expressed genes in the cerebellum, thalamus, and hypothalamus were selected based on the criteria of having a standardised value equal to or greater than 1.75.

### 2.2. Development of the ATLANTE Pipeline

The goal of this study was to find regions of the brain that might be functionally sensitive to deleterious mutations across several genes associated with MD. Given that there was no existing platform to explore this concept, a new analysis pipeline was created and named ATLANTE (Analysis Tool for Local Association of Neuronal Transcript Expression). Although the pipeline can be used to search for brain regions based on any user-generated list, its function is described here within the context of MD-associated genes since this represents a major motivator for its creation. Here, in the first application of the ATLANTE pipeline, regions of the brain that only shared a high expression of genes were the focus. The rationale for choosing to search for brain regions based on shared high expression levels was that high expression is easily definable and measurable. This is in contrast to low expression, which can be vulnerable to false negatives in cases of low, but non-negligible, expression levels. Moreover, genes that exhibit high expression (with low tolerability to mutation) have been shown to be particularly vulnerable to disease-causing *de novo* polymorphisms [41]. Since gene co-expression networks map to functional brain networks [42], the successful application of ATLANTE should identify functionally relevant regions of the brain.

The ATLANTE pipeline was created using python and operates in three major steps: (1) the generation of a reference ‘count’ of scores for highly expressed genes in discrete regions of the brain, (2) the scoring of these regions of the brain based on the high expression of a user-generated gene list, and, (3) comparing the outputs to detect statistically significant brain regions specific to the user-generated gene list.

The determination of high gene expression within individual brain regions by ATLANTE relies on data from the Human Protein Atlas version 24.0 and Ensembl version 109 (Courtesy of Human Protein Atlas, www.proteinatlas.org (accessed on 10 May 2025) [43]). The data cover 20,162 human genes and include transcript expression levels across 193 unique regions of the human brain. In this study, individual genes were not filtered out based on expected expression profiles, such as ‘no brain expression’ or ‘non-specific brain expression’, in order to avoid introducing unjustified bias. The mean and standard deviation of normalised mRNA expression transcripts per million (nTPM) in all 193 brain regions were determined for each gene. Brain regions were considered as having the ‘highest expression’ for any given gene if the nTPM value exceeded 1.96 standard deviations above the mean across all 193 brain regions for that gene. The resultant list of brain regions with the ‘highest expression’ for each gene was used as the input data for ATLANTE (below).

Reference counts were generated via iterative simulations that selected random genes from the ‘highest expression’ input data. For each iteration, X random genes were picked, where X was equal to the length of the user-generated gene list. Counts-per-region were produced by counting how many genes were most ‘highly expressed’ in each region. The process of the random selection and generation of counts-per-region was iterated 1,000,000 times to create a population that effectively represented the expected values of how many genes are most highly expressed in each brain region (for any given gene list of length X). Counts-per-region were then generated for the user-generated list, which, in this study, was the curated MD-associated gene list (described in Section 2.1; itemised in Table A1). In order to detect statistically significant brain regions, the mean and standard deviation (gene count per brain region) of the reference count population was calculated and a threshold of 1.96 standard deviations above the mean was set. The threshold was Bonferroni-corrected for multiple comparisons across the N = 193 brain regions, with each treated as an independent comparison. Any brain region that contained a number of genes that crossed the Bonferoni-corrected threshold was considered to be ‘enriched’ for highly expressed MD-associated genes. Fold-enrichment scores were generated for brain regions that crossed the threshold by dividing the MD-associated gene count by the mean gene count generated from the reference count process.

### 2.3. MD Gene-Region Network Graph Generation, Analysis, and Community Clustering

To explore the relationship between the MD-associated genes and brain regions identified by ATLANTE, a network graph was generated where nodes represented individual genes and edges represented a shared brain region wherein any two genes were both highly expressed. For this analysis, only genes and brain regions identified via application of the ATLANTE pipeline using the curated list of MD-associated genes were included, with 115 genes and 9 brain regions. A 115 × 9 gene–brain region matrix was generated where each cell/entry represented the presence or absence of a gene within a brain region (‘presence’ here means ‘highest expression’ in this brain region, as defined in Section 2.2). This matrix was used to assay each of the 6670 possible gene pairs for their shared presence within each identified brain region (0 for no shared presence; 1 for shared presence). The number of brain regions was summed if each gene pair shared more than one brain region. The resulting data were used to create a weighted network graph where each node represented a single gene, and each edge represented a shared brain region between genes (edge weight determined by the number of shared brain regions).

Nodal strength was calculated by summing all weighted edges for each node. Betweenness centrality was calculated by summing all the fractions of shortest path pairs passing through each node.

Clustering was performed on the gene-region network graph to identify emergent features that might indicate shared or independent molecular identities of the brain regions found in the enrichment analysis. A Louvain model [44] was used for clustering, with the goal of identifying large-scale, non-overlapping communities without any required assumptions about existing clusters. Although this method is less sensitive to small, granular community features, it introduces less bias and produces a more conservative outcome.

### 2.4. Cluster-Informed Gene Ontology Analysis

Gene lists based on Louvain clustering annotations were used as inputs for gene ontology (GO) analysis using ShinyGO version 0.82 [45]. A false discovery rate of 0.05 was set as the threshold to determine statistically significant terms relating to biological pathways. Outputs used in this study include false discovery rate and fold enrichment.

### 2.5. Statistical, Visualisation and Data Management Tools

All data handling, manipulation, analysis, and visualisation for this study was performed in Jupyter Notebook 7.2.2 [46]. Data handling and organisation was performed using *pandas* 2.2.3 [47]. Data visualisation was performed using Matplotlib 3.10 [48]. Graph theory analyses and data visualisation were performed using NetworkX 3.5 [49]. All Jupyter codes used in this study can be made available upon request.

## 3. Results

### 3.1. Creation and Validation of the ATLANTE Pipeline

In order to identify novel brain regions that may be associated with MD, this study tested whether MD-associated genes identified via GWAS meta-analyses display high levels of expression in shared locations across the brain. A custom analysis pipeline, called ATLANTE, was developed to search for brain regions in this way (see Section 2.2; Figure 1A). Before using ATLANTE for MD-associated genes, the pipeline required functional validation. To achieve this, ‘positive control’ gene lists were created that contained highly expressed genes from three separate brain regions: the cerebellum, the thalamus, and the hypothalamus (Figure 1B). These regions were selected because they represent well-defined and discrete parts of the brain that have been extensively mapped at a molecular level. BioGPS was used to access highly expressed genes in all three regions [38,39,40] and to develop custom gene lists. ATLANTE correctly identified brain regions associated with each of the three ‘positive control’ gene lists, including those within the cerebellum (Figure 1C), thalamus (Figure 1D), and hypothalamus (Figure 1E). For the list of genes highly expressed in the cerebellum, the results were highly consistent and accurate; all identified regions resided within the cerebellum, including the cerebellar cortex (CC), floculonodular lobe (FNL), and vermis (Ve). For the lists of genes highly expressed in the thalamus and hypothalamus, some non-specific regions were identified, such as the medial periolivary nuclei (MPO) for the thalamus gene list or the paramedian reticular nucleus (PMR) for the hypothalamus gene list. Although this may indicate a low rate of type 1 errors, it is possible that the regions share sufficient molecular identity to be co-identified from the same list of genes.

### 3.2. MD-Associated Genes Are Enriched in Discrete Brain Regions

Following successful validation, ATLANTE was used to identify brain regions based on MD-associated genes. A list of 268 high-confidence genes sourced from four recent GWAS meta-analyses of MD [22,29,30,32] was curated for use in this study. Application of the ATLANTE pipeline to the curated list of MD-associated genes identified nine brain regions that crossed the threshold for statistical significance when compared to the simulation-generated reference counts (Figure 2A,B). These brain regions included the temporal pole (TP), anterior transverse temporal gyrus (aTTG), olfactory tubercle (OT), ventral tegmental area (VTA), postcentral gyrus (PCG), CA1 region of the hippocampus (CA1), olfactory area (OA), perirhinal gyrus (PG), and posterior insular cortex (pIC). Thus, MD-associated genes are enriched in discrete brain regions, many of which cluster around the temporal lobe.

### 3.3. Brain Regions Exhibit Both Shared and Non-Overlapping MD-Associated Gene Architecture

Of the original 268 MD-associated genes used for the ATLANTE pipeline, 115 contributed to the identification of statistically significant brain regions. The median number of contributing genes per identified brain region was 20, which suggested some degree of shared MD-associated genetic signature between brain regions. To examine this further, a network graph was created wherein nodes represented genes and edges represented shared brain regions between genes (e.g., if gene A and gene B both contributed to enrichment of a brain region, they would share an edge; see Section 2.3).

The resultant network graph revealed unique brain region-based relationships between genes, with a mixture of highly integrated and less integrated node communities (Figure 3A). The existence of less integrated communities suggested that some brain regions might be defined by unique genetic architectures. To investigate this empirically, a clustering analysis was performed using a Louvain hierarchical approach, which identifies communities in an unsupervised manner, therefore introducing less bias [44]. The Louvain model successfully identified four independent clusters within the network (Figure 3A). Considering the underlying nine brain regions, the identification of four clusters suggested some shared MD-associated genetic architecture. To explore this further, each gene was classified based on its cluster identity and the cluster composition of each brain region was plotted by percentage. When visualised in this way, a clear association emerged between the Louvain clusters and the brain regions. Clusters 2, 3, and 4 largely comprised MD-associated genes enriched in the PCG, CA1, and VTA, respectively (Figure 3B). The remaining brain regions were instead defined by a significant proportion of cluster 1 genes intermixed with a small number of other clusters (Figure 3B). The pIC is notable in this group as it is defined entirely by cluster 1 genes. Thus, the PCG, CA1, and VTA have relatively unique MD-associated genetic signatures, whereas the TP, aTTG, OT, OA, PG, and pIC share extensive genetic overlap.

### 3.4. Hub Genes Identified via Nodal Analysis

Some nodes within the network graph appeared to hold special value as either hubs or bridges. These nodes might represent genes with unique value from a mechanistic or diagnostic perspective. A nodal analysis was performed to explore this via the quantitation of node properties.

To identify hubs, nodal strength was calculated for each node by summing all edges (including weights) and ordered from largest to smallest (Figure 4A). The gene with the highest nodal strength was PRKN, followed by CNTN5 and GRM5. PRKN was particularly interesting as a hub node since it appeared to share edges with nodes across multiple Louvain clusters (Figure 3A). Next, to identify bridge nodes, betweenness centrality was calculated by summing the fractions of pairs of shortest paths for each node and ordered from largest to smallest (Figure 4B). The gene with the highest betweenness centrality was DCC, followed by PRKN and CAMK1D. The identification of DCC as a bridge node appeared to result from its connection between the CA1-dominant cluster 3 and the VTA-dominant cluster 4 (Figure 3A). CAMK1D instead appeared to bridge the PCG-dominant cluster 2 and cluster 3, whereas PRKN appeared to bridge clusters 1, 2, and 3 (Figure 3A). These results thus identify genes of interest that may hold special relevance to MD pathology by impacting several highly relevant brain regions simultaneously. PRKN, in particular, was identified by both the nodal and betweenness centrality analyses, suggesting it may play an underappreciated role in MD.

### 3.5. Regional Localisation of Gene Ontology Signatures

The increased regional resolution and subsequent network clustering analyses performed in this study provided a unique opportunity to investigate the source of previously identified gene ontology (GO) signatures. Prior studies have reported that MD-associated genes are enriched for GO terms relating to neuronal differentiation and synaptic processes [26,32], but it is unclear whether the genes that contribute to these GO pathways are specifically associated with any particular region of the brain (via high expression, for example). To address this, gene lists specific to each of the network graph Louvain clusters (Figure 3A) were interrogated for statistically enriched biological pathways using GO analysis (see Section 2.4). The results for each cluster were notably distinct, suggesting specific regional localisation of GO signatures. Clusters 1 and 2 were most strongly associated with terms related to synapses (Figure 5A,B), which suggested that synaptic dysfunction may be particularly relevant to the brain regions associated with the genes in these clusters. Clusters 3 and 4 were instead more strongly associated with terms relating to developmental processes, with cluster 3 containing additional terms describing cell biological processes (such as mitophagy and lipid transport; Figure 5C) and cluster 4 featuring terms suggestive of ventricular and/or stem cell biology (Figure 5D). Thus, the GO biological processes of MD-associated genes segregate according to the cluster communities identified in this study.

## 4. Discussion

### 4.1. Summary of Results

In this study, a new analysis pipeline called ATLANTE was developed to identify brain regions that are uniquely enriched for MD-associated genes. The pipeline operates by combining information pertaining to user-generated gene lists (which, for this particular study, was a curated list of MD-associated genes previously identified via GWAS meta-analyses [22,29,30,32]) with regional brain tissue expression data from the Human Protein Atlas [43]. Notably, though the Human Protein Atlas provides high regional coverage across the brain, it is limited to protein coding genes. As such, non-protein coding sequences and pseudogenes were not included in the analysis. Though this may skew the outputs of ATLANTE, the roles of non-protein coding genes and pseudogenes in disease remain incompletely elucidated. Additional work is required to build more extensive catalogues of these types of transcripts across the human brain. Until then, the approach used by ATLANTE is arguably the most conservative and relevant to the known biological mechanisms of gene dysregulation in disease.

Application of the ATLANTE pipeline in this study successfully identified nine discrete brain regions, including the temporal pole (TP), anterior transverse temporal gyrus (aTTG), olfactory tubercle (OT), ventral tegmental area (VTA), postcentral gyrus (PCG), CA1 of the hippocampus (CA1), olfactory area (OA), perirhinal gyrus (PG), and posterior insular cortex (pIC). Network graph and clustering analyses of the relationships between the identified genes and brain regions revealed that the PCG, CA1, and VTA harbour largely unique MD-associated genetic signatures, whereas the TP, aTTG, OT, OA, PG, and pIC share significant overlap. Node-based analyses identified PRKN as a highly interconnected gene node that may serve as a candidate for follow-up study with respect to its contributions to the underlying mechanisms driving MD or as a clinically relevant marker of MD.

### 4.2. Major Depression Genes Associate with Dopaminergic Signalling, Olfaction, and Parkinson’s Disease

In this study, the VTA was identified as a brain region of interest. Identification of the VTA represents an important convergence between the results of this study and previously identified mechanisms underlying MD. The VTA is associated with reward/motivation and is characterised by a high density of dopaminergic neurons. Dysfunction of dopamine signalling has long been considered an important mechanism in MD, particularly as it is associated with anhedonia, which can impact motivation, decision-making, and overall quality of life [50]. Depressed or suicidal individuals exhibit reduced D2 dopamine receptor availability in the basal ganglia [51], enhanced D2/3 dopamine receptor availability in the amygdala [52], and reduced DAT dopamine transporter availability [53]. Moreover, high resolution 7-Tesla MRI scans have recently revealed that the VTA displays marked structural changes in patients with MD compared to healthy controls [54]. This congruence between the results of this study and a previously identified mechanism of MD pathobiology contributes to validating the usefulness of ATLANTE and reinforces the importance of dopaminergic signalling in MD. The identification of the OT and CA1 in combination with the VTA is particularly relevant in this regard, since both are output targets of the VTA [55,56], with the OT being a component of the mesolimbic reward pathway as a part of the ventral striatum [57].

The OA and the OT, specifically in the context of olfaction, also represent areas that have a known strong link to MD and thus represent another convergence between the findings of this study and the pre-existing literature on MD. Dysfunction in olfactory acuity has been observed in patients with MD [58] and the hedonic valuation of odorants is often shifted towards displeasure in depressed individuals compared to healthy controls [59]. The relationship between olfactory dysfunction and depression is proportional, such that increased hyposmia predicts increased MD symptom severity [60]. Olfactory dysfunction is, in fact, predictive of MD symptom development 5 or 10 years before onset, which is indicative of a complex bidirectionality regarding the relationship between olfaction and MD [61].

An additional convergence between the findings of this study and previous work comes in the form of established connections between MD, olfactory dysfunction, and Parkinson’s disease (PD) [62,63]. Indeed, up to 90% of PD patients experience hyposmia [64], oftentimes as an early and even preclinical non-motor symptom of disease [65,66]. This connection is particularly relevant considering the identification of olfactory brain regions in this study. The identification of PRKN as a major hub gene in this study, as well as the enrichment of a GO term for mitophagy, is intriguing to consider in the context of potential links between olfaction, MD, and PD. Indeed, it serves as additional evidence for a mechanistically close relationship between MD and PD. Additional existing lines of evidence that support this idea include the following: (1) depression is a common non-motor symptom of PD [67], (2) diagnosis of depression in individuals over the age of 50 increases the risk of developing PD [68], (3) bi-allelic mutations in PRKN cause early-onset PD [69], and (4) family members of individuals with early-onset PD carrying monoallelic PRKN mutations have a significantly elevated risk of experiencing depression compared with family members without PRKN mutations [70]. Furthermore, MD and PD share many similarities at the level of the cell and molecular mechanisms of disease. Both involve dysregulation of the dopamine neurotransmitter system [71,72], have life stress as a risk factor [73,74], are associated with detectable alterations in α-synuclein metabolism [75,76], and converge on mitochondrial dysfunction as a cellular hallmark of disease [77,78]. Given the extensive overlap between olfactory dysfunction, PD, and MD, consideration of shared underlying mechanisms may help develop new clinical biomarkers, such as advanced olfactory bulb testing or circulating markers relating to PRKN.

### 4.3. Cortical Regions Amenable to Clinical Screening and the Potential for MD Subtype Segregation

This study revealed several brain regions enriched for the high expression of MD-associated genes that are amenable to clinical assessment using electroencephalogram and transcranial stimulation. Cortical regions such as the pIC, PCG, TP, and aTTG have already been associated with MD [79,80,81,82,83,84,85,86]. These regions thus represent potentially useful sources of electrodiagnostic biomarkers for the advanced detection of early subclinical depression or to identify MD’s clinical subtypes. Despite receiving relatively little attention in the context of depression, the aTTG is a particularly interesting target because of the relatively specific role it plays. Also known as Heschl’s gyrus, the aTTG is part of the primary auditory cortex, wherein it contributes to processing acoustic features relating to music and speech [87,88,89]. It responds very specifically to spontaneous inner speech and not task-related inner speech [90], meaning it is closely involved in ruminative processes. A major aspect of MD is negative rumination (brooding), which has been recently identified as being more strongly symptomatically represented in the melancholic subtype of MD [91]. Future studies directed at the aTTG using clinically relevant screening tools may therefore lead to more accurate methods for the segregation of MD subtypes, specifically in the context of melancholia, and consequently improve accuracy in administering precision therapies. Additional clinical relevance from this study, specifically with respect to subtyping MD, might be gained by investigating the structure or function of the brain regions identified here in patients with different MD subtypes to test whether they segregate in any meaningful manner. Alternatively, ATLANTE could be applied to gene lists generated from more specific groups of patients pre-segregated based on MD subtype. Additional advances in gene sequencing technologies are required for this, however, given that the most recent and successful genome-wide association studies required pooling large numbers of individuals with a more generalised definition of MD in order to detect rare polymorphisms.

The lack of identified frontal lobe regions in this study is notable given the well-documented evidence of frontal lobe involvement in MD, particularly in the form of asymmetry of electroencephalogram alpha-band activity [92,93,94]. This association is based upon hypoactivation of the left frontal lobe and the accompanying hyperactivation of the right frontal lobe, which correspond with behavioural withdrawal, an underlying factor in depressive behaviour [95,96]. The lack of identified frontal lobe regions in this study has two possible explanations. The first is that the choices made in developing the three major steps of the ATLANTE pipeline introduced bias(es) that selectively impacted sensitivity in detecting frontal lobe regions. It is possible that genes associated with MD that affect the frontal lobe do so via largely different mechanisms compared to the genes that impact the brain regions identified in this study. Searching for regional enrichment based on high gene expression is one example of a bias that may have been introduced, because this selectively ignores brain regions with finely tuned or low expression. These regions may be especially sensitive to changes in gene dosage (which can be particularly true for genes encoding transcription factors), and thus may be strongly affected by mutations that lead to haploinsufficiency [97]. Indeed, it is possible that MD arises due to imbalances in brain cell function across multiple regions, driven by combinations of molecular loss- and gain-of-function. Mapping regions of low MD-associated gene expression is thus of high value; it might be possible to achieve this through integrating growing databases from new technologies, such as spatial transcriptomics, into the ATLANTE pipeline. The second possibility is that frontal lobe dysfunction may be a secondary, reactive, feature of MD, following primary dysfunctions in the regions identified in this study. Many of the regions identified here are closely related to regulation of emotional, reward and sensory information. These regions may be arguably more fundamental to the determination of mood than regions in the frontal lobe, which are more closely related to executive control and decision-making (in response to emotional, reward, and sensory information). More studies are needed to discern which of these possibilities best explains the results of this study.

### 4.4. Gene Ontology-by-Cluster Indiciates the Presence of Molecularly Distinct Pathologies

Gene ontology (GO) analysis is regularly performed on lists of genes identified by genome-wide association studies. Although this is often informative, the lists are limited by a lack of biological context, which may preclude the functional grouping of genes based on discrete pathological processes. The application of ATLANTE provides an avenue to group genes into biologically relevant clusters to search for more specific pathways related to the pathology of a disease. Indeed, in this study, network clustering of MD-associated genes based on their spatial relationships in the brain yielded discrete groups (four clusters) that were amenable to further GO analysis. Cluster 1 most closely replicated previously identified GO analysis results [26,32], with many terms relating to synapse biology. Cluster 2 produced terms associated with the electrical regulation of cardiac cell function. It is relatively well-known that GO database terms are not ideally suited to brain cell biology, and although some tools are currently in development to enhance GO analysis for gene function in brain cells, most remain limited [98]. As such, one interpretation of this result is that cluster 2 genes relate to regulation of the electrical properties of neurons or astrocytes, rather than cardiac cells specifically. Clusters 1 and 2 thus share some functional overlap in the sense that they regulate the functioning, and possibly the plasticity or reactivity, of mature brain cells. It is interesting to note here that clusters 1 and 2 mostly comprise genes highly expressed in cortical brain regions, indicating a shared mechanism of cortical dysfunction in MD. The GO terms for clusters 3 and 4, on the other hand, appeared more relevant to developing cells. The GO of both clusters yielded terms relevant to multipotent cells, cellular differentiation, and processes relating to cellular projection, which is most relevant to neurons during development. Since cluster 3 and 4 genes are mostly expressed in the hippocampus and ventral tegmental area, respectively, this might suggest that the limbic system is particularly susceptible to specific developmental architectures that enhance the susceptibility of individuals to MD. It is interesting to consider whether these unique clusters represent processes that are differentially relevant to disparate subtypes of MD or, instead, disparate pathological steps involved in generating an individual’s risk to the development of MD overall. More work is required to clarify this.

## 5. Conclusions

This study generated a novel analysis pipeline to search for brain regions associated with Major Depression using high-confidence genes sourced from recent genome-wide association studies. The new analysis pipeline, called ATLANTE, identified nine brain regions, including the temporal pole, anterior transverse temporal gyrus, olfactory tubercle, ventral tegmental area, postcentral gyrus, CA1 of the hippocampus, olfactory area, perirhinal gyrus, and posterior insular cortex. These brain regions represent novel targets for advanced clinical screening of depression, specifically in the context of the early subclinical screening and detection of disease subtypes. Network graph analyses identified the PRKN gene as a key player across multiple identified brain regions, reinforcing the connection between Major Depression and Parkinson’s disease. Clinical assays targeting processes relevant to the molecular functions of PRKN might provide new avenues for the early or more accurate detection of depression.

## Figures and Tables

**Figure 1 neurolint-17-00096-f001:**
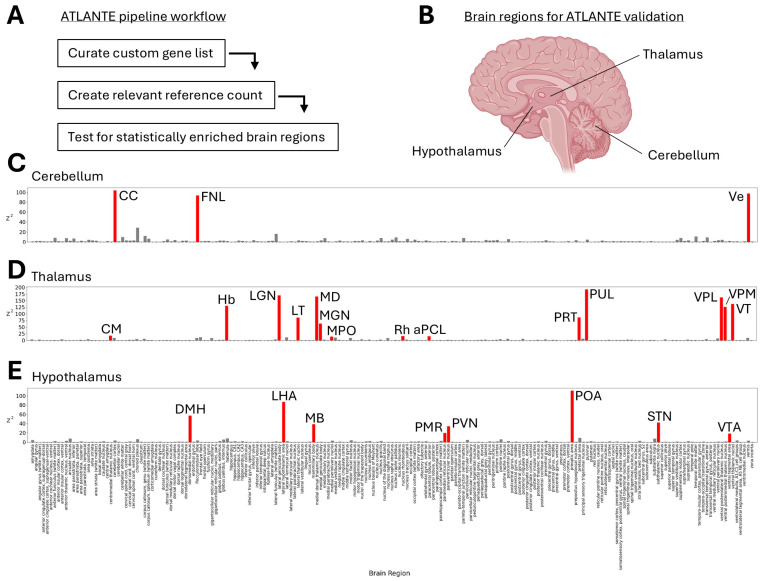
Development of the ATLANTE pipeline. (**A**) The workflow of ATLANTE contains three major steps: (1) curation of a custom gene list, (2) creation of a reference count for all brain regions, and (3) testing to identify statsitically significant brain regions. (**B**) Diagrammatic representation of the three brain regions used as positive controls to validate ATLANTE. (**C**–**E**) Bar plots of results for the list of genes highly expressed in the cerebellum (**C**), thalamus (**D**), and hypothalamus (**E**). Values shown are squared z-scores (Z^2^), with grey bars representing brain regions that did not reach statistical significance and red bars representing brain regions that did reach statistical significance. CC—cerebellar cortex; FNL—folocculonodular lobe; Ve—vermis; CM—centromedial thalamic nucleus; Hb—habenula; LGN—lateral geniculate nucleus; LT—lateral thalamic nuclei; MD—medial dorsal thalamic nucleus; MGN—medial geniculate body; MPO—medial periolivary nucleus; Rh—nucleus rhomboideus; aPCL—anterior paracentral lobule; PRT—pretectal area; PUL—pulvinar; VPL—ventral posterolateral thalamic nucleus; VPM—ventral posteromedial thalamic nucleus; VT—ventral thalamic nuclei; DMH—dorsomedial nucleus; LHA—lateral hypothalamic nucleus; MB—mammillary body; PMR—paramedian reticular nucleus; PVN—paraventricular nucleus; POA—preoptic area; STN—subthalamic nucleus; VTA—ventral tegmental area.

**Figure 2 neurolint-17-00096-f002:**
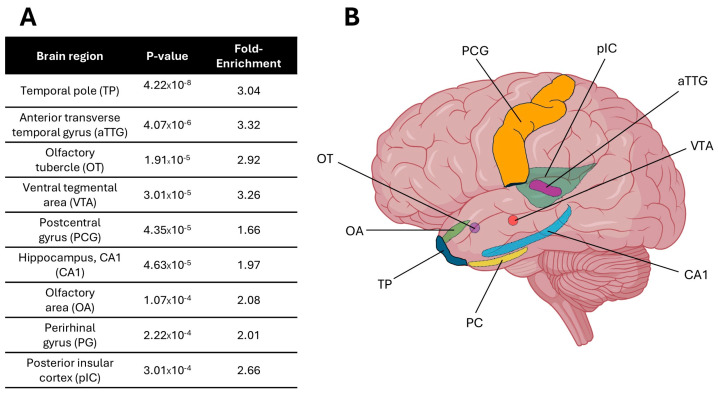
Enrichment of Major Depression (MD)-associated transcripts in discrete brain regions. (**A**) Brain regions that crossed the threshold for statistical significance, ordered by *p*-value. ‘Fold-Enrichment’ refers to the how many times more MD-associated genes were enriched in the brain region relative to expected frequencies for an equal number of randomly selected genes (MD gene count/reference gene count). (**B**) Diagrammatic representation of the brain regions identified in this study. TP = temporal pole; aTTG = anterior transverse temporal gyrus; OT = olfactory tubercle; VTA = ventral tegmental area; PCG = postcentral gyrus; CA1 = CA1 region of the hippocampus; OA = olfactory area; PG = perirhinal gyrus; pIC = posterior insular cortex.

**Figure 3 neurolint-17-00096-f003:**
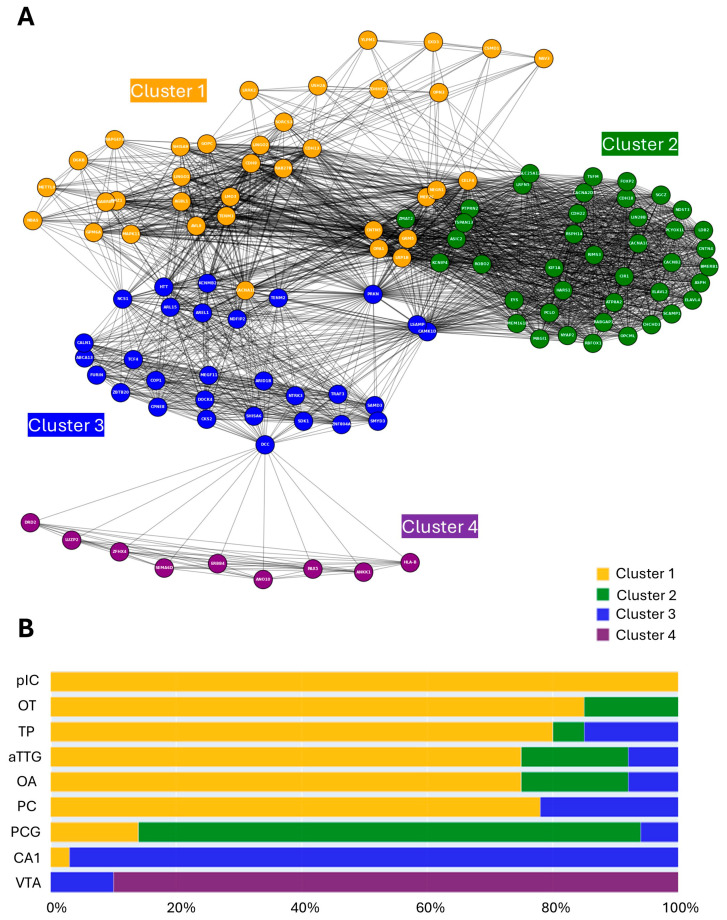
Distinct and shared genetic architectures of identified brain regions. (**A**) Network graph and clustering analysis of MD-associated genes and brain regions identified in this study. Nodes represent individual genes. Edges represent shared high expression of a pair of genes (nodes) within a brain region. Line thickness is directly proportional to the number of brain regions shared by a pair of genes (nodes). Nodes are colour-coded according to the results of Louvain clustering. Cluster identities are indicated in the colour of each cluster. (**B**) Bar graph showing the composition by cluster of each brain region identified in this study. Each gene was assigned an identity based on its cluster and the percentage of each cluster was calculated for each brain region. TP—temporal pole; aTTG—anterior transverse temporal gyrus; OT—olfactory tubercle; VTA—ventral tegmental area; PCG—postcentral gyrus; CA1—CA1 region of the hippocampus; OA—olfactory area; PG—perirhinal gyrus; pIC—posterior insular cortex.

**Figure 4 neurolint-17-00096-f004:**
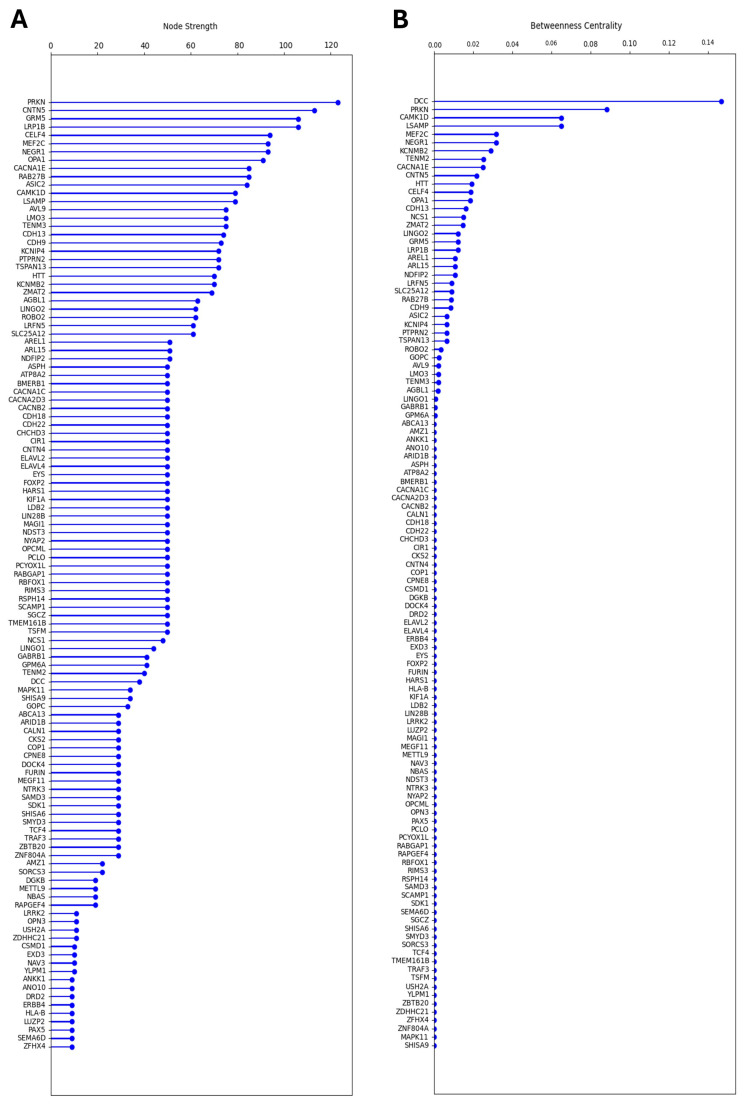
Nodal analyses for each node (gene) in the network graph. (**A**) Bar graph depicting results for the nodal strength analysis. Genes are listed in order of nodal strength. (**B**) Bar graph depicting results for the betweenness centrality analysis. Gene are listed in order of betweenness centrality.

**Figure 5 neurolint-17-00096-f005:**
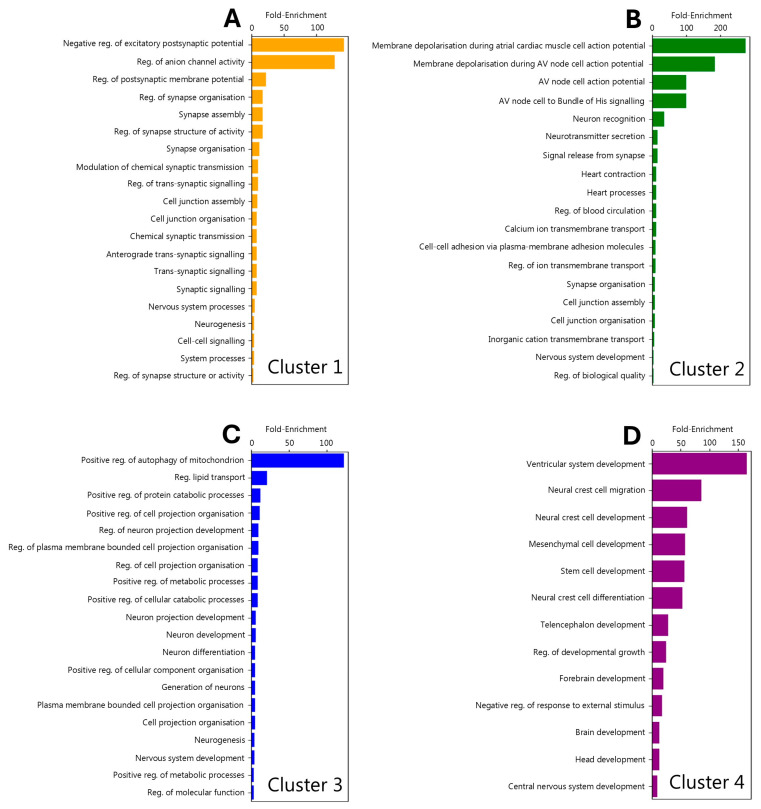
Gene ontology (GO) analysis of gene lists based on network graph Louvain clusters. (**A**–**D**) Bar graphs displaying fold enrichment score for all biological pathway GO terms that passed the false discovery rate threshold for cluster 1 (**A**), cluster 2 (**B**), cluster 3 (**C**), and cluster 4 (**D**). Terms are organised from largest to smallest by fold enrichment score.

## Data Availability

The original contributions presented in this study are included in the article. Further inquiries can be directed to the corresponding author(s).

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
