# Peer review of "Novel Gene-Informed Regional Brain Targets for Clinical Screening for Major Depression"

_2035-8377, 2025, doi:10.3390/neurolint17060096_

Round 1

Reviewer 1 Report

Comments and Suggestions for Authors

The paper by Odierna et al. describes development of a novel computational method applied to the detection of brain regions that are enriched for expression of risk genes associated with major depression (MD). The authors identified 9 brain regions where MD risk genes are richly expressed and used network analysis to detect 4 clusters of genes. These clusters were further analyzed to identify hub genes, such as PRKN and DCC, and to search for shared gene ontology (GO) terms. The studies were well designed, appropriately controlled and provided novel findings. The paper might be further strengthened by addressing the comments below.

  1. The authors used ATLANTE to identify brain regions with high expression of MD risk genes. It seems ATLANTE can also be used to determine brain areas with inordinately low gene expression although this may be more challenging and subject to false negatives. Perhaps, the current analysis missed prefrontal regions strongly implicated in MD by brain imaging and other approaches because these regions have low rather than high expression of the risk genes. These areas would then be vulnerable owing to less buffering by gene dosage and might be more sensitive to loss-of-function mutations or small decreases in gene expression. Although there is brief mention of this type of possibility, additional discussion of this likely explanation would round out this aspect of the paper. It would be logical if MD results from an imbalance in brain function with overactive limbic areas with high gene expression experiencing faulty top-down regulation by prefrontal areas owing to low expression of some of the same genes.
  2. The GO by Cluster analysis was very interesting and merits further comment in the Discussion. Although many interpretations of their findings are possible, one could argue that genes in the different clusters cause disease-related phenotypes at different stages of brain development/function. Clusters 4 and 3 would appear to affect early stages of brain development as compared to Clusters 1 and 2 that affect mature neurons and specialized functions such as synapses. Thus, the effects of Clusters 1 and 2 might not fully manifest until the brain has matured and could include modulation of the gene products by learning and chronic behavioral states. This could go along with the authors’ idea that prefrontal areas were missed because these regions are affected secondarily in the disease process. In any event, it would help to hear more of the authors’ thinking about these novel findings.
  3. Minor comment: reference 28 is incomplete.

Author Response

Reviewer 1

Comment 1: The authors used ATLANTE to identify brain regions with high expression of MD risk genes. It seems ATLANTE can also be used to determine brain areas with inordinately low gene expression although this may be more challenging and subject to false negatives. Perhaps, the current analysis missed prefrontal regions strongly implicated in MD by brain imaging and other approaches because these regions have low rather than high expression of the risk genes. These areas would then be vulnerable owing to less buffering by gene dosage and might be more sensitive to loss-of-function mutations or small decreases in gene expression. Although there is brief mention of this type of possibility, additional discussion of this likely explanation would round out this aspect of the paper. It would be logical if MD results from an imbalance in brain function with overactive limbic areas with high gene expression experiencing faulty top-down regulation by prefrontal areas owing to low expression of some of the same genes.

Response 1: We agree with the reviewer’s interpretation that there are likely regions in the brain that share low expression of MD-associated genes and that these regions would be rather vulnerable to loss of gene function, even in response to small fluctuations in expression. As the reviewer correctly identifies, determination of low expression is challenging and subject to false negatives. Indeed, we discuss these concepts on p3, lines 105-110, and p14, lines 420-426. We have added some text to explore the concept even further. Please find the added text on p14, lines 426-430.

Comment 2: The GO by Cluster analysis was very interesting and merits further comment in the Discussion. Although many interpretations of their findings are possible, one could argue that genes in the different clusters cause disease-related phenotypes at different stages of brain development/function. Clusters 4 and 3 would appear to affect early stages of brain development as compared to Clusters 1 and 2 that affect mature neurons and specialized functions such as synapses. Thus, the effects of Clusters 1 and 2 might not fully manifest until the brain has matured and could include modulation of the gene products by learning and chronic behavioral states. This could go along with the authors’ idea that prefrontal areas were missed because these regions are affected secondarily in the disease process. In any event, it would help to hear more of the authors’ thinking about these novel findings.

Response 2: The reviewer is correct that the GO analysis by cluster is novel and was not appropriately discussed in our paper. To address this, we have added a new section in the discussion. This new section can be found on p14-p15, lines 437-462.

Comment 3: Minor comment: reference 28 is incomplete.

Response 3: We thank the reviewer for identifying the incomplete reference. Please find the correct citation on line 535.

Reviewer 2 Report

Comments and Suggestions for Authors

In this manuscript, the authors generate a new analysis pipeline (ATLANTE) that enables the association of gene expression patterns with specific brain regions in major depression (MD). Using ATLANTE, the authors identified nine brain regions implicated in MD. Furthermore, they conducted network and clustering analysis and highlighted a list of MD-associated genes, with particular emphasis on PRKN, which may play a vital role in the disorder.

Comments:

  1. In the introduction, the authors acknowledge the involvement of a substantial number of non-coding RNAs and pseudogenes in MD. However, their analysis focuses on protein-coding genes, which may introduce bias. To address this limitation, the authors should consider incorporating additional analyses using non-coding RNA datasets or addressing the impact in the discussion part.
  2. Although the study identifies nine brain regions enriched for MD-associated genes, it does not establish a clear correspondence between these regions and clinically defined MD subtypes, as suggested in the abstract and introduction. This gap should be addressed in the discussion to clarify the clinical relevance of the findings.
  3. The authors identified PRKN as a gene associated with Parkinson’s disease (PD) and MD. The authors should discuss the potential mechanistic overlap between MD and PD.

Author Response

Reviewer 2

Comment 1: In the introduction, the authors acknowledge the involvement of a substantial number of non-coding RNAs and pseudogenes in MD. However, their analysis focuses on protein-coding genes, which may introduce bias. To address this limitation, the authors should consider incorporating additional analyses using non-coding RNA datasets or addressing the impact in the discussion part.

Response 1: The reviewer is correct that non-protein coding RNAs and pseudogenes have been identified in genome wide association studies on individuals with MD, as well as transcript-focused approaches. Unfortunately, databases that include reports of non-protein coding transcripts and pseudogenes at a similar spatial resolution across the brain as that provided by the Human Protein Atlas for coding genes do not currently exist. Moreover, the exact role of these types of transcripts in the pathophysiology of MD remains incompletely understood. We have added discussion addressing this limitation in the discussion. Please find the new text on p12, lines 328-335.

Comment 2: Although the study identifies nine brain regions enriched for MD-associated genes, it does not establish a clear correspondence between these regions and clinically defined MD subtypes, as suggested in the abstract and introduction. This gap should be addressed in the discussion to clarify the clinical relevance of the findings.

Response 2: We thank the reviewer for identifying that additional discussion is required with respect to how the results of our paper relate to clinical subtypes of MD. We have added new discussion of this on p13-p14, lines 405-412. Please note that p13, lines 497-404, discuss the possibility of utilising the anterior transverse temporal gyrus as a specific region of interest for the melancholic subtype of depression.

Comment 3: The authors identified PRKN as a gene associated with Parkinson’s disease (PD) and MD. The authors should discuss the potential mechanistic overlap between MD and PD.

Response 3: We have added some commentary on the mechanistic connection between MD and PD. This new addition to the discussion can be found on p13, lines 383-387. Please note that connections between PD and MD are also discussed on p13, lines 374-383.

Round 2

Reviewer 2 Report

Comments and Suggestions for Authors

The authors have addressed my comments.